# Analysis of Injuries and Wellness in Blind Athletes during an International Football Competition

**DOI:** 10.3390/ijerph19148827

**Published:** 2022-07-20

**Authors:** Jesús Muñoz-Jiménez, Luisa Gámez-Calvo, Daniel Rojas-Valverde, Kiko León, José M. Gamonales

**Affiliations:** 1Facultad de Ciencias del Deporte, Universidad de Extremadura, 10005 Cáceres, Spain; lgamezna@alumnos.unex.es (L.G.-C.); fleon@unex.es (K.L.); martingamonales@unex.es (J.M.G.); 2Núcleo de Estudios en Alto Rendimiento y Salud (NARS-CIDISAD), Escuela Ciencias del Movimiento Humano y Calidad de Vida (CIEMHCAVI), Universidad Nacional, Heredia 86-3000, Costa Rica; 3Clínica de Lesiones Deportivas (Rehab&Readapt), Escuela Ciencias del Movimiento Humano y Calidad de Vida (CIEMHCAVI), Universidad Nacional, Heredia 86-3000, Costa Rica; 4Facultad de Ciencias de la Salud, Universidad Francisco de Vitoria, 28223 Madrid, Spain

**Keywords:** 5-a-side football, sport technology, injuries, pain perception, load indexes

## Abstract

Five-a-side football for blind people is the only adapted football modality present at the Paralympic games. Fa5 is a collaborative-opposition sport in which its participants play with no vision, which causes numerous impacts. At the London 2012 Paralympic Games, it was the sport with the highest incidence of sports injuries. The main objective of this work is to analyse the association between pain perception; spatio-temporal, mechanical, and metabolic workload with injuries; and wellness in players during an international Fa5 competition. The following variables, monitored during an International Fa5 Tournament, were analysed: general well-being, perception of pain and injuries, and the spatio-temporal and metabolic workload. The results show that the incidence of injuries increases as the tournament progresses, where injured players reported greater muscle pain and stress before the matches started. Besides, the players’ internal and external load did not explain the incidence of injury. Still, the values obtained in the wellness questionnaire, the perception of pain, and stress suggested they contributed to the incidence of injury.

## 1. Introduction

Football is the world’s most popular sport [1], with more than 265 million practitioners [2]. Football is a sport that allows adaptation to any type of collective, regardless of its characteristics [3]. At present, football is the sport with a more significant number of adapted modalities [4]. Athletes are functionally classified to play under equal conditions in competitive adapted football modalities, depending on the type and degree of disability. These classifications are ruled by the different national and international federations [5,6,7].

The only adapted football modality present in the Paralympic Games is five-a-side football (Fa5), whose regulations are ruled by the International Federation of Associated Football (FIFA) [8]. Fa5 has been one of the most striking Paralympic sports, followed by spectators since its first participation in Athens in 2004 [9]. Fa5 is a collaborative-opposition collective sport played on a 20 × 40 m rectangular outdoor field to the correct acoustics [10]. The playing area is surrounded by side barriers, which would enable the game’s continuity. Fa5 teams comprise five players; one goalkeeper (without visual impairment), and four blind outfield players. During the competitions, the players of both teams, except for the goalkeepers, all wear a mask to provide similar conditions [9,11]. One of the most significant adaptations of Fa5 is that the ball has been modified to make a jingling sound, which allows players to know their location during the game [12].

Although football is one of the most practised and studied sports, the evidence related to Fa5 is not common in scientific literature. Fa5 studies focus on this modality’s historical context and tactics game [13]. In addition, concern about players’ injuries and teams’ rising interest related to their prevention has increased [14]. In recent years, advances in technology and research have allowed the use of new technological devices to quantify the external load of athletes [15]. The uses of the beneficial devices increased in recent years due to their portability and a large number of recorded variables. Specifically, inertial devices provide information about the magnitude and frequency of actions based on accelerometry (e.g., impacts, jumps, changes of direction) and actions based on speed (e.g., distance covered, maximum speed, speed mean) and related to indicators of internal loads, such as heart rate [15]. Data analysis can help identify internal and external variables and contextual factors that influence performance [15]. The investigations on these variables are usually related to indicators of internal loads, such as heart rate, to precisely characterise each sport’s physical and physiological demands [16,17,18,19].

Some studies carried out in adapted football modalities, such as football 7 for people with cerebral palsy [16,20] or the Fa5 [15], have used inertial devices, and heart rate monitors to determine the internal and external load variables that influence the performance of the players. In Fa5, few pieces of scientific literature analyse internal and external workload demands. A recent study showed that players generally had more accelerations per minute when losing due to an increase in the effort to reclaim the game. In addition, the offensive players who scored goals usually covered a greater distance and at a higher speed than the rest of their teammates [15].

The current investigations that analyse the injuries of athletes with disabilities do not usually focus specifically on Fa5 players. However, Fa5 at the London 2012 Paralympic Games had the highest incidence of sports injuries [12]. This may be due to the high load and intensity of training, the game’s characteristics, the players’ qualities [21], and the reduced size of the field [22]. In addition to injuries in the lower body, the appearance of abrasions and lacerations is widespread [21]. An injury is understood as damage that occurs in the body. It’s a general term that refers to the damage caused by accidents, falls, blows, burns, weapons, and other causes [21]. Pain and pain perception is highly linked to sports experience and performance [23]. Despite the attention paid to sports injuries and pain by the coaching staff of teams and athletes, the perception of pain and its importance for athletes’ daily life and physical performance has not been studied [24].

This article aims to provide insights in addressing Fa5 collaborative-opposition sport, a Paralympic 5-a-side football event for blind competitors. Given the lack of attention paid to the perception of pain and its importance for athletes’ daily life and physical performance, the authors intend to analyse the association between space–time and mechanical load with injuries and wellness in blind players during an international Fa5 competition.

## 2. Materials and Methods

### 2.1. Design

This section describes in detail the methodology used in the research development.

Spatio-temporal, mechanical, and metabolic workload and wellness variables of ten blind football players of the Spanish senior national team were monitored during the blind football international tournament held in Sevilla, Spain, in 2019. International representative teams of the Czech Republic, Andalusia, Italy, and Spain took part in the three-days-congested fixture tournament. Spanish Fa5 team played against adversaries in a congested tournament playing one match per day.

The spatio-temporal, mechanical, and metabolic workload variables were clustered using exploratory factor analysis. Finally, the association between the grouped (factors) variables was associated with players’ wellness and injury incidence.

### 2.2. Participants

Data were collected from ten top-international male blind football players (29.20 ± 10.9 years; 178.40 ± 7.48 cm; and 72.42 ± 7.04 kg) from the Spanish senior national team (two goalkeepers, three defenders, five forwards), who participated in the international tournament, held in Sevilla. The descriptive analysis of the sample is outlined in Table 1.

Only those participants playing at least 20 min (2/3 of the match) of the matches were included in the analysis. Before data collection, all participants signed an informed consent document, which contained all the investigation details. Moreover, the study was approved by the International Paralympic Committee (IPC), the Institutional Review Board (Universidad de Extremadura, Reg. Code 67/2017), and the team players. Furthermore, the teams ‘staff and tournament managers consented to participate in this research.

### 2.3. Materials and Procedures

#### 2.3.1. Spatio-Temporal, Mechanical, and Metabolic Workload

A commercial inertial sensor (WIMU PRO^TM^, Real Track Systems, Almeria, Spain) containing four accelerometers, a gyroscope, and a magnetometer recorded mechanical data. A GNSS captures the positioning and location of the sensors using trilateration via a constellation of satellites as a reference. The devices were placed in the inter-scapulae line at the level of T2–T4 vertebrae and were attached using a tight-fitting vest to reduce motion artifact signal [15,25]. In fact, static high-intensity actions without covering ground (jumps, collisions, falls, tackles, etc.) cannot be recorded by time-motion systems. Still, they can be measured with high accuracy by accelerometers [25]. Collation and analysis of data from studies reporting temporal changes in the occurrence of higher intensity accelerations and decelerations during competitive match play may help acquire knowledge regarding the magnitude of the decline and potential impact that this may have on match performance and injury risk. In particular, intense accelerations and decelerations could be especially susceptible to neuromuscular fatigue and consequently to an exacerbated risk of sustaining injury [26]. Moreover, heart rate as a metabolic indicator was assessed using ANT+ technology linked with the inertial sensors using a chest-adjusted cardiac monitor (HRM3, Garmin^TM^, Olathe, KS, USA).

#### 2.3.2. Wellness and Injury Incidence

The football players were instructed to complete a customized perceived wellness questionnaire. The participants were familiarized with the questionnaire in advance. The authors assessed the players’ general well-being, fatigue, and the presence of pain during the competition through a wellness validated questionnaire review sheet based on athletes monitoring recommendations [27,28]. Each of the players was asked to provide details based on the components of self-perceived tools used to assess players wellness. A five-point Likert-scale questionnaire with a range of 5–lowest/1–highest that has scaled (i) players’ level of fatigue (highly fatigued to not fatigued at all), (ii) sleep quality (hardly slept to great sleep), (iii) muscle soreness (extremely sore to not sore at all), (iv) stress levels (highly stressed), and (v) mood (highly irritable to very positive mood). Injury incidence components are shown for each of the ten players in Table 2. Players reported wellness conditions before each match. Each section score was reported, and the total recorded score was analysed.

An injury was considered when signs and symptoms referred to damage occurred in the body structure causing dysfunction in relation to soccer. In this study, two different kinds of injuries were registered: contusion, understood as the result of direct impact to the muscle causing a compression against a hard surface (e.g., the bone); and strain, when the muscle fibre is overstretched or torn.

#### 2.3.3. Statistical Analysis

All the data were expressed in mean and standard deviation. The normality of the data was confirmed using the Shapiro wilk test. The variables were selected among maximum, average values of time-related data to compare between matches. Moreover, all raw data were scaled and centred using Z-scores to avoid individual variation patterns and potential fluctuation of measures. A z-score must be understood as the expression of a score in the number of standard deviation units the raw score is above or below the distribution mean. It is calculated as follows [27]:$$Z-\mathrm{score}=\frac{\mathrm{player}\;\mathrm{score}-\mathrm{player}\;\mathrm{average}}{\mathrm{player}\;\mathrm{standar}\;\mathrm{deviation}}$$

A mixed analysis of variance (MANOVA) was performed to examine the main effects for the match-injured players by match (1st, 2nd, and 3rd) in wellness-related variables and external load variables. Post-Hoch analysis was performed using Bonferroni´s correction within time course changes and wellness scores. The MANOVA was chosen considering the data contains within and between participant’s variables. Omega squared (*ω_p_*^2^) was used to qualify and quantify the magnitude of the differences (effect size: ES) as follows: <0.01 trivial; >0.01 small; >0.06 moderate and >0.14 large [29].

Spatio-temporal and mechanical variables were clustered using the most used explorative factor analysis technique, the principal component analysis (PCA), following methodological and practical guidelines for sports science research [30]. Variables reporting variance = 0 were excluded. PCA suitability was confirmed through Kaiser–Meyer–Olkin (KMO) values (KMO = 0.62 − 0.67) and Bartlett sphericity test significance (*p* < 0.05) [31]. Those eigenvalues >1 were considered for the extraction of each PC, and an orthogonal rotation method (VariMax) was selected to identify high correlations between components to offer different information [30]. Those PC loadings >0.6 were considered for extraction, and only the highest factor loading was retained when a cross-loading was found between PCs [31]. All PCA data were reported using specific sports science research protocols [32].

The cumulative binomial probability of getting injured was calculated. It is understood as the sum of the probabilities for all events from 0 to x [33]. This probability of injury incidence was estimated using the cumulative distribution function of the binomial distribution. It is what is needed to compute the probability of observing less than or more than a certain number of outcomes (e.g., injury) from a number of trials (e.g., three matches). The binomial cumulative distribution function was estimated using the following formulae:$$F\left(x\right)=\mathrm{Pr}\left[X\leq x\right]=\overset{x}{\underset{i=0}{\sum }}f\left(i\right)$$

Finally, a binomial logistic regression was performed to analyse how each PC extracted explains the wellness score. The PCs were introduced and analysed using a backward elimination method. Data fitting in the logistic regression model were confirmed using *Chi*^2^ (*R*^2^). Moreover, model results were reported using *R*^2^, *β*,*_exp_β*, and 95% *IC*. Alpha was set at *p* < 0.05, and all data were analysed and systematised using the Statistical Package for the Social Sciences (SPSS, IBM, SPSS Statistics, v.22.0 Chicago, IL, USA).

## 3. Results

### 3.1. Wellness Scores by Match and Injury Incidence

As shown in Table 2, the incidence of injury was 0% for the first match, 50% for the second match, and 90% for the third match. Consequently, based on the cumulative distribution function, the cumulative binomial probability of getting injured was 14.57%.

Clustering spatio-temporal and mechanical variables through an explorative PC (principal component) analysis—previously testing their sampling adequacy through a Kaiser–Meyer–Olkin (KMO) measure and Bartlett sphericity test significance—the authors have found very large to nearly perfect higher levels of muscle pain (ES = 0.14, large) and feeling stressed (ES: 0.4, large) (Table 3) correlations, while both the stress and mood gradually decreased as the tournament progressed. This means the players are reported to be more relaxed as the tournament progresses.

### 3.2. Spatio-Temporal Workload by Match and Injury Incidence

There were no differences in spatio-temporal variables by match and injury incidence, as shown in Table 4. Instead, very large differences were found in scaled and centred values in mechanical workload variables, as shown in Table 5.

### 3.3. Mechanical Workload Variables by Match and Injury Incidence

Finally, the results obtained concerning the differences by match and injury incidence throughout the tournament in mechanical variables are shown in Table 5. Injured players performed higher accelerations and decelerations (ES = 0.56, large) and fewer steps (ES = 0.09, moderate).

### 3.4. Principal Component Analysis

To interpret each principal component, the authors have examined the magnitude and direction of the coefficients for the original variables in Figure 1. Two PCs were extracted and grouped based on the variables’ characteristics of each of the PCs. After PCA of spatio-temporal and mechanical workload variables were performed, the model explained the total variance by 87.99% and 87.61%, respectively (see loadings and grouping in Figure 1).

Binomial logistic regression results indicate that both PC´s factors of spatio-temporal workload variables explain injuries incidence in 3% of cases (*R^2^* = *0.03,*
*β* = −0.12, *_exp_**β* = 0.89, and 95% *IC* = 0.42:1.87). Additionally, the mechanical workload variables explained the incidence of injury in 3% of cases (*R ^2^*= *0.03,*
*β* = −0.46, *_exp_**β* = 0.63, and 95% *IC* = 0.23:1.77). None of the regression models were significant despite the high relative percentage of explanation of the incidence of injury. Contrarily, perceived muscle soreness and stress level explained injury incidence significantly (0.046) in 27.1% of cases (*R^2^* = *0.27,*
*β* = −0.64, *_exp_**β* = 0.53, and 95% *IC* = 0.27:1.05). Therefore, the larger the absolute value of the coefficient speed and distance, the more important the corresponding variable is in calculating the component. Whereas the mechanical workload variables explained the common incidence of injury, perceived muscle soreness and stress level explained injury incidence significantly.

## 4. Discussion

From a descriptive point of view, it is observed that the incidence of injuries increases as the tournament progresses; this incidence being 0% in the first match, 50% in the second match, and 90% in the last match. As in previous studies [12], it may be due to the accumulation of impacts. The tournament is developed in a single weekend, where three games are played, which may compromise the correct physical recovery of the players. Despite the absence of studies focused on the incidence of injuries in Fa5 players, the results of this study are like those carried out in the London 2012 Paralympic Games [12], where it is concluded that Fa5 is the Paralympic sport with the highest incidence of injuries of all those analysed. Most of the injuries recorded in the Seville tournament occurred in the upper body, specifically in the trunk (69.4%), and were caused by contusions (84.61%), which differs from what was stated by De Campos et al., 2015 [34], who concluded that the predominant lesions occur in the lower limbs. Therefore, the authors strengthen how Fa5 confirms itself as a Paralympic sport with the highest incidence of injuries of all those analysed, with injuries especially in the upper body and lesions in the lower limbs. Pain and the perception of pain are highly linked to sports experience and performance, and then they negatively affect the performance of athletes. It was found that the probability of getting injured during this tournament was 14.57%.

The results show significant differences in muscle pain and stress perception, with injured players being those with higher levels in both variables. However, in both injured and non-injured players, the mood improves throughout the tournament. This may be conditioned by the good results obtained by the team.

The results show large differences in pain perception and stress levels among the injured players, who presented higher levels of muscle pain (ES = 0.14) and stress (ES = 0.4) than the non-injured players. The characteristics of the sports context of Fa5 determine the presence of injuries due to the high number of impacts produced between players [12], a consequence of the actions with and without the ball made in a game situation [35], and the high intensity of the play [22] in the absence of view. Pain and the perception of pain are highly linked to sports experience and performance [23], so competing with pain can negatively affect the performance of athletes.

The PCA provides key information about factors that explain the variance of the spatio-temporal variables (distance, explosive distance, maximum speed and average speed) and the mechanical variables of workload (acceleration, decelerations, impacts, steps, and player load). The maximum speed reached the spatiotemporal workload variables per game, and the average speed explained 66.79% of the total variance. In comparison, distance and explosive distance explain 21.20% of the total variance, the principal components of this set of variables. Regarding the mechanical variables of workload, impacts, steps, and workload per player explain 49.43% of the total variance, while accelerations and decelerations explain 38.18% of the total variance. These variables define the difference between values obtained by different players.

Concerning the mechanical variables, the results show that injured players performed a more significant number of accelerations and decelerations, with fewer steps, than non-injured players. The external load variables, the distance covered, and the maximum and average speed reached do not explain the incidence of injuries. Contrarily, the non-blinded football players usually performed actions less quick (30% slower) [36], with respect to non-blinded players. Blinded players usually performed a greater number of strides and trunk flexion angle at the lumbar level compared to non-blinded players [37]. This could explain why blinded players could present a reduced likelihood of getting injured, due to loading magnitude-related injuries [38]. In blinded players, the running related injuries may result from cumulate microtrauma provoked by magnitude (e.g., vertical ground reaction forces) and number (e.g., higher number of strides) of actions [36].

However, the variables of perception of pain and stress suggest a 27% contribution to the incidence of injury. This way, it can be understood that the subjective values of pain perception, general wellness, and fatigue can predict a higher injury incidence. The perception of pain and stress results coincide with those presented in other works [39] that affirm that these values are affected by training and competition. The authors find other large differences in pain perception and stress levels among the injured players, who presented higher muscle pain and stress levels than the non-injured players. Values are higher than expected, also considering that in a non-Paralympic football match, the workload resulting from lumbar segments will likely be higher than those assessed in thoracic segments. 

### Limitations

While this study presents some evidence on the incidence of injuries and the relation to wellness perception in blind players. Moreover, some evidence regarding how the external load explains the injury rate in these findings must be seen considering some limitations. Due to the special characteristics of the participants (e.g., blinded soccer players), this is a quasi-experimental paper with the significant limitation that the sample is rather small, and data are somewhat restricted with only three data collection points after different matches from the same ten players.

## 5. Conclusions

This research paper concludes that the incidence of injuries increases as the tournament progresses; therefore, allowing enough time between matches to permit the correct recovery of players and reduce the incidence of injuries is essential. Furthermore, injured players report greater muscle pain and stress before the start of the game. On the other hand, the players’ internal and external load did not explain the injury incidence. Still, the values obtained in the wellness questionnaire and the perception of pain and stress suggest contributing to this injury incidence.

Although not explicitly discussed in this study, it seems that the result of the match can influence the mood and the feeling of relaxation of the players since these values were better throughout the tournament after winning all the matches. Moreover, blind athletes’ ability to anticipate collisions can be improved. Therefore, they must be prepared to reduce the magnitude of the impacts during training and matches.

This is a quasi-experimental paper due to the lack of scientific articles published on Fa5, with the significant limitation that the sample is rather small, and data are somewhat restricted with only three data collection points after different matches from the same ten players.

## Figures and Tables

**Figure 1 ijerph-19-08827-f001:**
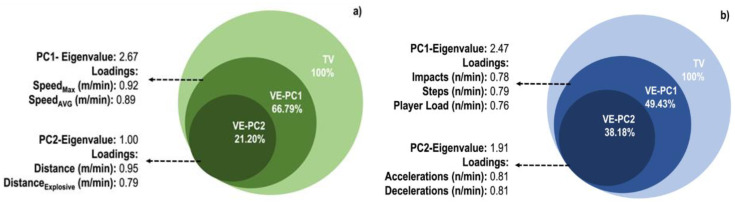
Principal components analysis clustering representation of (**a**) spatio-temporal and (**b**) mechanical workload variables. TV: total variance, VE: variance explained, PC: principal component.

**Table 1 ijerph-19-08827-t001:** Descriptive analysis of the sample.

ID	Height (cm)	Weight (kg)	Body Mass Index	Position	Playing Time by Match			
					1st Match	2nd Match	3rd Match	Total
1	186.00	78.50	21.10	Goalkeeper	0:30:00	0:30:00	0:30:00	1:30:00
2	175.00	71.00	20.28	Goalkeeper	0:30:00	0:30:00	0:30:00	1:30:00
3	163.00	59.60	18.28	Forward	0:45:01	0:47:01	0:31:01	2:03:03
4	180.00	80.60	22.39	Defender	0:14:59	0:14:59	0:29:00	0:58:58
5	179.00	72.00	20.11	Defender	0:24:16	0:29:16	0:24:20	1:17:52
6	172.00	69.00	20.05	Defender	0:15:40	0:13:40	0:35:40	1:05:00
7	187.00	80.00	21.39	Forward	0:44:20	0:44:20	0:34:20	2:03:00
8	187.00	80.00	21.39	Forward	0:35:55	0:30:55	0:25:55	1:32:45
9	179.00	72.00	20.11	Forward	0:39:44	0:31:44	0:31:44	1:43:12
10	176.00	66.00	18.75	Forward	0:26:05	0:29:05	0:28:15	1:23:25

**Table 2 ijerph-19-08827-t002:** Incidence and injury type by match.

Player ID	1st Match		2nd Match		3rd Match	
	Location	Type	Location	Type	Location	Type
1	-	-	-	-	-	-
2	-	-	-	-	Right leg	Strain
3	-	-	Left leg	Strain	Torso	Contusion
4	-	-	Torso	Contusion	Right leg	Contusion
5	-	-	Torso	Contusion	Torso	Contusion
6	-	-	Torso	Contusion	Left leg	Contusion
7	-	-	-	-	Torso	Contusion
8	-	-	-	-	Torso	Contusion
9	-	-	-	-	Torso	Contusion
10	-	-	-	-	Torso	Contusion

**Table 3 ijerph-19-08827-t003:** Raw data of wellness scores by match and injury incidence.

Variable	Injury Incidence	1st Match	2nd Match	3rd Match	F_Match-effect_ (*p*-Value)	*ω_p_*^2^ (Rating)
Fatigue	Non-injured (n = 16)	3.10 ± 0.32	3.20 ± 0.45	3.00 ± 0.00	0.88 (0.43)	0 (trivial)
	Injured (n = 14)	0.00 ± 0.00	3.20 ± 0.84	2.33 ± 1.12		
F_Injury-effect_ (*p*-value)		0.51 (0.48)			F_Interaction_ (*p*-value)	ω_p_^2^ (rating)
ω_p_^2^ (rating)		0 (trivial)			0.51 (0.48)	0 (trivial)
Sleep quality	Non-injured (n = 16)	3.20 ± 1.23	3.00 ± 1.23	4.00 ± 0.00	0.11 (0.89)	0 (trivial)
	Injured (n = 14)	0.00 ± 0.00	3.20 ± 1.10	2.78 ± 1.10		
F_Injury-effect_ (*p*-value)		0.51 (0.48)			F_Interaction_ (*p*-value)	ω_p_^2^ (rating)
ω_p_^2^ (rating)		0 (trivial)			0.99 (0.33)	0 (trivial)
Muscle soreness	Non-injured (n = 16)	3.90 ± 1.29	4.00 ± 1.00	5.00 ± 0.00	0.28 (0.76)	0 (trivial)
	Injured (n = 14)	0.00 ± 0.00	3.40 ± 0.90	2.67 ± 1.66		
F_Injury-effect_ (*p*-value)		3.23 (0.08)			F_Interaction_ (*p*-value)	ω_p_^2^ (rating)
ω_p_^2^ (rating)		0.14 (large)			1.13 (0.30)	0 (trivial)
Stress	Non-injured (n = 16)	3.50 ± 0.71	3.60 ± 0.55	5.00 ± 0.00	2.58 (0.1)	0.06 (small)
	Injured (n = 14)	0.00 ± 0.00	2.80 ± 0.45	3.11 ± 0.78		
F_Injury-effect_ (*p*-value)		10.50 (<0.01)			F_Interaction_(*p*-value)	ω_p_^2^ (rating)
ω_p_^2^ (rating)		0.40 (large)			1.73 (0.20)	0.03 (small)
Mood	Non-injured (n = 16)	4.30 ± 0.48	4.60 ± 0.55	5.00 ± 0.00	1.57 (0.23)	0.02 (small)
	Injured (n = 14)	0.00 ± 0.00	4.80 ± 0.45	4.89 ± 0.33		
F_Injury-effect_ (*p*-value)		0.03 (0.87)			F_Interaction_ (*p*-value)	ω_p_^2^ (rating)
ω_p_^2^ (rating)		0 (trivial)			0.32 (0.58)	0 (trivial)

Note: Raw data were presented as a reference, but results of Z-scores were analysed.

**Table 4 ijerph-19-08827-t004:** Raw data of spatio-temporal workload variables by match and injury incidence.

Variable	Injury Incidence	1st Match	2nd Match	3rd Match	F_Match-effect_ (*p*-Value)	*ω_p_*^2^ (Rating)
Distance (m/min)	Non-injured	44.08 ± 22.74	40.61 ± 27.29	51.27 ± 0.00	0.87 (0.43)	0 (trivial)
	Injured	0.00 ± 0.00	30.67 ± 17.19	50.15 ± 14.16		
F_Injury-effect_ (*p*-value)		0.19 (0.66)			F_Interaction_ (*p*-value)	ω_p_^2^ (rating)
ω_p_^2^ (rating)		0 (trivial)			0.12 (0.73)	0 (trivial)
Distance _Explosive_ (m/min)	Non-injured	5.35 ± 4.06	5.07 ± 2.95	4.88 ± 0.00	0.63 (0.54)	0 (trivial)
	Injured	0.00 ± 0.00	2.82 ± 3.30	6.35 ± 1.92		
F_Injury-effect_ (*p*-value)		0.04 (0.84)			F_Interaction_ (*p*-value)	ω_p_^2^ (rating)
ω_p_^2^ (rating)		0 (trivial)			1 (0.35)	0 (trivial)
Speed_Max_ (km/h)	Non-injured	16.27 ± 4.56	15.34 ± 4.67	14.46 ± 0.00	0.15 (0.86)	0 (trivial)
	Injured	0.00 ± 0.00	16.02 ± 5.67	17.02 ± 4.31		
F_Injury-effect_ (*p*-value)		0.32 (0.58)			F_Interaction_ (*p*-value)	ω_p_^2^ (rating)
ω_p_^2^ (rating)		0 (trivial)			0.11 (0.75)	0 (trivial)
Speed_Avg_ (km/h)	Non-injured	3.17 ± 0.85	3.08 ± 0.79	2.61 ± 0.00	0.36 (0.70)	0 (trivial)
	Injured	0.00 ± 0.00	2.99 ± 0.82	3.61 ± 0.76		
F_Injury-effect_ (*p*-value)		0.86 (0.36)			F_Interaction_ (*p*-value)	ω_p_^2^ (rating)
ω_p_^2^ (rating)		0 (trivial)			1.19 (0.29)	0 (trivial)

Note: Raw data were presented as a reference, but results of Z-scores were analysed.

**Table 5 ijerph-19-08827-t005:** Raw data of mechanical workload variables by match and injury incidence.

Variable	Injury Incidence	1st Match	2nd Match	3rd Match	F_Match-effect_ (*p*-Value)	*ω_p_*^2^ (Rating)
Accelerations (n/min)	Non-injured	36.26 ± 8.08	65.39 ± 18.52	39.94 ± 0.00	18.81 (<0.01)	0.56 (large)
	Injured	0.00 ± 0.00	72.68 ± 20.58	39.63 ± 4.32		
F_Injury-effect_ (*p*-value)		0.21 (0.65)			F_Interaction_ (*p* value)	ω_p_^2^ (rating)
ω_p_^2^ (rating)		0 (trivial)			0.25 (0.62)	0 (trivial)
Decelerations (n/min)	Non-injured	36.24 ± 8.02	65.52 ± 18.64	39.97 ± 0.00	18.62 (<0.01)	0.56 (large)
	Injured	0.00 ± 0.00	72.59 ± 20.77	39.63 ± 4.32		
F_Injury-effect_ (*p*-value)		0.19 (0.66)			F_Interaction_ (*p*-value)	ω_p_^2^ (rating)
ω_p_^2^ (rating)		0 (trivial)			0.24 (0.63)	0 (trivial)
Impacts (n/min)	Non-injured	20.57 ± 11.31	17.39 ± 9.78	5.53 ± 0.00	0.92 (0.41)	0 (trivial)
	Injured	0.00 ± 0.00	12.53 ± 7.28	23.07 ± 15.03		
F_Injury-effect_ (*p*-value)		0.75 (0.39)			F_Interaction_ (*p*-value)	ω_p_^2^ (rating)
ω_p_^2^ (rating)		0 (trivial)			2.34 (0.14)	
Steps (n/min)	Non-injured	39.70 ± 17.80	25.77 ± 9.85	41.02 ± 0.00	3.82 (0.04)	0.09 (moderate)
	Injured	0.00 ± 0.00	24.54 ± 13.75	43.67 ± 9.49		
F_Injury-effect_ (*p*-value)		0 (0.99)			F_Interaction_ (*p*-value)	ω_p_^2^ (rating)
ω_p_^2^ (rating)		0 (trivial)			0.09 (0.77)	0 (trivial)
Player Load (a.u./min)	Non-injured	0.54 ± 0.21	0.59 ± 0.15	0.58± 0.00	0.13 (0.88)	0 (trivial)
	Injured	0.00 ± 0.00	0.50 ± 0.18	0.62 ± 0.19		
F_Injury-effect_ (*p*-value)		0.06 (0.81)			F_Interaction_ (*p*-value)	ω_p_^2^ (rating)
ω_p_^2^ (rating)		0 (trivial)			0.34 (0.57)	0 (trivial)

Note: Raw data were presented as a reference, but results of Z-scores were analysed.

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
