# Peer review of "Analysis of Injuries and Wellness in Blind Athletes during an International Football Competition"

_ijerph, 2022, doi:10.3390/ijerph19148827_

Round 1
Reviewer 1 Report
In their revised version of the paper the authors intend to analyse the association between:
- pain perception,
- Spatio-temporal, mechanical and metabolic workload
with
- Injuries, and
- wellness
in players during an international Fa5 competition.
As cleared in previous discussions with the authors, Fa5 is collaborative-opposition sport, a paralympic 5-a-side football for blind competitors (see my previous REVIEW_IJERPH-1632538).
Section 1. Introduction, it is an explanation of Fa5’s features in the landscape of international sports competitions. Being practiced by blind competitors, this sport includes however several internal and external load variables to influence the performance of blind players (see authors’ line 65). The authors then explicitly would enhance the perception of pain in these athletes’ daily life to detect better situations and potential safety interventions about their physical performance (see authors’ line 80).
Table 1 at line 106, provides a first descriptive analysis of the ten athletes (sampled because of the authors’ sought association) related to their physical features (i. height, ii. weight, iii. body mass) and the playing time (minimum total participation of the sampled athletes during the three single matches is of 20 minutes) by the three matches played in Sevilla during an International Tournament (three days; one match per day). Playing time of the ten athletes is shown by the three single matches’ partial values and in total value by sum of the three considered partial times.
Three data sources have therefore been collected on the same 10 players.
Through a kind of FACIT-F (Functional Assessment of Chronic Illness Therapy - Fatigue) questionnaire, the authors then have realized a non-probability (convenience) respondent-driven sampling (RDS) of these 10 players.
The sampling frame should be of 66 players in total, because a match is played by two teams, each with a maximum of eleven players.
Then, Three matches = 66 players.
It is a kind of capture-recapture sampling (repeated counting data) without the chance of the McLaughlin Method [1]—since respondents are always the same 10 people—to get the overlap. Because of a questionable model fit (non-heterogenous detection probabilities), McLaughlin [2] would be (10 * 10): 10 = 10, that it seems rather silly because catchability of population is equal, and therefore the population size is the same as the sample size.
Sub-paragraph 2.3 Materials and procedures and followings give information about the way authors have calculated the Spatio-temporal, mechanical and metabolic workload of the ten sampled athletes: i.e., accelerometers helpful in collating and analysing of data with “temporal changes in the occurrence of higher intensity accelerations and decelerations during competitive match play […]” (see authors’ lines 124-128).
HR (heart rate) variable was also assessed through another specific technology.
The authors make precede their sought association by a five-point Likert-scale questionnaire submitted to the ten sampled athletes before of each three matches in the Sevilla International Tournament. Through this kind of FACIT-F (Functional Assessment of Chronic Illness Therapy - Fatigue) questionnaire they have drawn a total recorded score of:
(i) players’ level of fatigue,
(ii) sleep quality,
(iii) muscle soreness,
(iv) stress levels, and
(v) mood.
with range 5 – lowest/1 – highest (see authors’ line 139).
This scale would serve to authors to highlight how the percentage of injury incidence components such as “contusion” and “strain” (see authors’ Table 2 at line 187) is able to cause “dysfunction in relation to soccer,” (see authors’ lines 143-144) leave aside “the perception of pain and its importance for athletes' daily life and physical performance […]” (see authors’ lines 78-80).
Considering that the authors targets to set association between physical injuries, location and type of injury and other physical players variables (such as weight and body mass) and wellness (principally considered as muscle pain and stress before matches, I believe it would be important to highlight in the paper:
· The probability of success (i.e., getting a Injury) on any single trial is 13/30 0.433333
· The number of trials is 30 (because the 10 players have been considered for
the three matches) 30
· The number of successes is 13 (since we define “getting a Injury” as success) 13
Injuries are in fact present in 4 cases for the 2nd match and in 9 cases in the 3rd match (see Table 2 at line 186 by authors).
A «cumulative distribution function (CDF) of the Binomial distribution is what is needed when you need to compute the probability of observing less than or more than a certain number of events/outcomes/successes from a number of trials. The Binomial CDF formula is simple:
Therefore, the cumulative binomial probability is simply the sum of the probabilities for all events from 0 to x.» [3].
In this case: the Probability of exactly 13 events is of 0.14573667110237 (14.57%).
So, the ten sampled players during the three matches have a probability of 14.57% to get injured.
At a first examination data would indicate that in the three matches percentages of some crucial variables may be different notwithstanding the ten players are always the same.
|
|
|
||||||||||||||||||||||||||||||||||||||||||||||||||||||||||||||||||||||||||||||||||||||||
|
Note: values are the partial % in the three matches pondered by the totals of % in the three matches. |
||||||||||||||||||||||||||||||||||||||||||||||||||||||||||||||||||||||||||||||||||||||||||
Here below the data related to the ten players in the three matches:
Table 1_REVIEW: RANDOM CREATION OF DATA FOR THE THREE MATCHES.
|
|
||||||||||||||||||||||||||||||||||||||||||||||||||||||||||||||||||||||||||||||||||||||||||||||||||||||||||||||||||||||||||||||||||||||||||||||||||||||||||||||||||||||||||||||||||||||||||||||||||||||||||||||||||||||||||||||||||||||||||||||||||||||||||||||||||||||||||||||||||||||||||||||||||||||||||||||||||||||||||||||||||||||||||||||||||||||||||||||||||||||||||||||||||||||||||||||||||||||||||||||||||||||||||||||||||||||||||||||||||||||||||||||||||||||||||||||||||||||||||||||||||||||||||||||||||||||||||||||||||||||||||||||||||||||||||||||||||||||||||||||||||||||||||||||||||||||||||||||||||||||||||||||||||||||||||||||||||||||||||||||||||||||||||||||||||||||||||||||||||||||||||||||||||||||||||||||||||||||||||||||||||||||||||||||||||||||||||||||||||||||||||||||||||||||||||||||||||||||||||||||||||||||||||||||||||||||||||||||||||||
|
Note: Values are generated with a RANDOM formula in excel and respect the range of values shown by authors in Tables 2, 3, 4, and 5. Important: the non-injured/injured (NO, 0; YES, 1) values in this Table are RANDOMLY calculated and therefore do not respect the 13 events OF “Injury” as highlighted above in the main body of the review. |
||||||||||||||||||||||||||||||||||||||||||||||||||||||||||||||||||||||||||||||||||||||||||||||||||||||||||||||||||||||||||||||||||||||||||||||||||||||||||||||||||||||||||||||||||||||||||||||||||||||||||||||||||||||||||||||||||||||||||||||||||||||||||||||||||||||||||||||||||||||||||||||||||||||||||||||||||||||||||||||||||||||||||||||||||||||||||||||||||||||||||||||||||||||||||||||||||||||||||||||||||||||||||||||||||||||||||||||||||||||||||||||||||||||||||||||||||||||||||||||||||||||||||||||||||||||||||||||||||||||||||||||||||||||||||||||||||||||||||||||||||||||||||||||||||||||||||||||||||||||||||||||||||||||||||||||||||||||||||||||||||||||||||||||||||||||||||||||||||||||||||||||||||||||||||||||||||||||||||||||||||||||||||||||||||||||||||||||||||||||||||||||||||||||||||||||||||||||||||||||||||||||||||||||||||||||||||||||||||||
Then, I have proved a moving average calculation, that is a technical indicator that investors and traders use to determine the trend direction of securities. It is calculated by adding up all the data points during a specific period and dividing the sum by the number of time periods. This indicator sums up the data points of a financial security over a specific time period and divides the total by the number of data points to arrive at an average. It is called a “moving” average because it is continually recalculated based on the latest price data.
I have applied the SMA (Simple Moving Average) formula to the variables related to the research—except for the physical features (i. height, ii. weight, iii. body mass), the playing time (minimum total participation of the sampled athletes during the three single matches is of 20 minutes; average in total is of 1 hour, 34 minutes), and the presence, type and location of injury that were categorical variables and rather fixed along the three matches.
The formula for Simple Moving Average is written as follows:
SMA = (A1 + A2 + ……….An) / n
Where:
- A is the average in period n
- n is the number of periods
Data had been previously coded according to the below graph:
|
|
|
The graph is created under the suggestions of PLS-Graph software, available at: https://swmath.org/software/11498. |
Here below the merging of all three matches variables to get the SMAs for each column:
Table 2_REVIEW: MERGING OF SCORES FOR THE VARIABLES TO GET SMAs (Simple Moving Averages).
|
||||||||||||||||||||||||||||||||||||||||||||||||||||||||||||||||||||||||||||||||||||||||||||||||||||||||||||||||||||||||||||||||||||||||||||||||||||||||||||||||||||||||||||||||||||||||||||||||||||||||||||||||||||||||||||||||||||||||||||||||||||||||||||||||||||||||||||||||||||||||||||||||||||||||||||||||||||||||||||||||||||||||||||||||||||||||||||||||||||||||||||||||||||||||||||||||||||||||||||||||||||||||||||||||||||||||||||||||||||||||||||||||
|
Note: Values are now rounded to two decimals. |
Slightly differently from authors, who have used of Z-scores to amalgamate data, here below the SMAs:
Table 3_REVIEW: SMAs OF AUTHORS’ RANDOMLY CREATED VARIABLES FOR THE REVIEW.
|
||||||||||||||||||||||||||||||||||||||||||||||||||||||||||||||||||||||||||||||||||||||||||||||||||||||||||||||||||||||||||||||||||||||||||||||||||||||||||||||||||||||||||||||||||||||||||||||||||||||||||||||||||||||||||||||||||||||||||||||||||||||||||||||||||||||||||||||||||||||||||||||||||||||||||||||||||||||||||||||||||||||||||||||||||||||||||||||||||||||||||||||||||||||||||||||||||||||||||||||||||||||||||||||||||||||||||||||||||||||||||||||||||||||||||||||||||||||||||||||||||||||||||||||||||||||||||||||||||||||||||||||||||||||||||||||||||||||||||||||||||||||||||||||||||||||||||||||||||||||||||||||||||||||||||||||||||||||||||||||||||||||||||||||||||||
|
|
Sub-paragraph 2.3.3 Statistical Analysis, explains all scores have been centred using Z-scores: that means the authors have provided the Spatio-temporal, mechanical and metabolic workload variables calculated through accelerometers, and the scores obtained in the five-point Likert-scale questionnaire submitted before of each three matches through their respective difference between players’ score and players’ average score divided by players’ standard deviation scores (see authors’ formula expressed at line 156).
Then, wellness-related variables (measured by the questionnaire) and external load variables (measured by the accelerometers) have been analysed through a mixed analysis of variance (MANOVA), together with post-hoc analysis and “Bonferroni’s correction within time course changes and wellness scores.” (see authors’ lines 158-164).
Rojas-Valverde, D., Pino-Ortega, J., Gómez-Carmona, C. D., & Rico-González, M. A Systematic Review of Methods and Criteria Standard Proposal for the Use of Principal Component Analysis in Team’s Sports Science (2020) has been a reference for the principal component analysis (PCA) with which authors have clustered Spatio-temporal and mechanical workload variables. Only the highest factor loading was retained when a cross-loading was found between PCs, according to the reference the authors mention Kaiser, H. F. The Application of Electronic Computers to Factor Analysis (1960).
Setting alpha at p < 0.05, the authors perform a SPSS binomial logistic regression after a previous backward elimination method to extract workload variables. This procedure is confirmed using the Chi2 (R2). See lines 165-181. As reported by the questionnaire, the incidence of injury was 0% for the first match, 50% for the second match, and 90% for the third match. See lines 184-186.
In Table 3 at line 195, the authors present raw data of wellness scores by match and injury incidence and find very large to nearly perfect ES (Effect Size) of muscle pain and feeling stressed among the ten sampled athletes—direct correlation—while the stress’ and mood’s seriousness both gradually decrease as the tournament progresses. In Table 4 at line 200, the authors present raw data of Spatio-temporal workload variables by match and injury incidence and find no differences if not the data are scaled and centred (as in Table 5 at line 206 where differences are very large).
Most of the injuries by the ten sampled athletes in Sevilla were on the upper body, especially in the trunk, differently from other authors’ findings as De Campos et al. (2015), who concluded that this kind of sports involve particularly lesions in the lower limbs. See authors’ lines 235-241.
Because of high incidence of injuries in both trunk and in lower limbs, “Fa5 confirms itself as a Paralympic sport with the highest incidence of injuries of all those analysed,” and the authors’ analysis may be also a way to prevent players’ collisions in future tournaments.
…………………………………………..
I think SMAs of values respect more the temporal shifting during the three matches for the same 10 players. Observed Injury’s variables (as well as their location & type) are better explained through this kind of bootstrap method allow to get 28 different observations’ sets of variables. Probability of injuries happening is fixed to 14.57%, value that could be useful for comparison of other tournaments’ alike happenings.
Many observations can therefore be derived without the risk people does not understand from whom data are observed. People are always the same 10 people in the three different matches, but the sampling basin is of 28 observations’ sets of variables for the same 10 players (see Table 3_REVIEW: SMAs OF VARIABLES).
I have named the ten players to know them (!), and below the Microsoft tool has calculated of these 28 observations’ sets of variables with replacement which new average of authors’ calculated variables is foreseeable for each of the ten players:
|
Row Labels |
Average of WE5 SMA |
Average of WE1 Simple Moving Average (SMA) |
|
ZHANG |
3.44 |
1.66 |
|
ALBERT |
3.015 |
2.77 |
|
JEFFREY |
2.855 |
3.2925 |
|
JO |
2.7425 |
2.93 |
|
MARC |
2.72 |
2.085 |
|
YUWEI |
2.703333333 |
3.48 |
|
TOM |
2.596666667 |
2.633333333 |
|
ROBERT |
2.0675 |
1.695 |
|
JACK |
2.0075 |
2.475 |
|
CHARLIE |
1.713333333 |
2.47 |
In Table 3_REVIEW: SMAs OF AUTHORS’ RANDOMLY CREATED VARIABLES FOR THE REVIEW, in fact names are sometime repeated because they are created again RANDOMLY among 10 chosen by me names [TOM; ALBERT; JO; ZHANG; MARC; CHARLIE; JACK; ROBERT; YUWEI; JEFFREY] along the 28 sets. Here above a couple of Microsoft examples of outcome about the variables expressed in simple moving averages.
…………………………………………..
Comments: A friend of mine hearing from me to speak about this sport under review asked to me: “Which special tools help these blind players to move during the tournament?”
The authors, in their first reply to our reviews, told me these blind athletes have no special needs neither impairment to play soccer, save the crude fact they are blind.
I should perhaps suppose they are less quick respect to ordinary soccer players. A renowned paper by [4] use of a probabilistic model to tell the audience ten blind male subjects who have a decreasing running speed have a reduced likelihood for stress fracture. “Loading magnitude” rather than “loading exposure,” has been also a factor of reducing risk in this kind of physical stress to players.
In addition, [5] have demonstrated that “Blindfolded football players present[ed] a shorter stride length, which consequently reduces running speed.” Then, this paper’s results regarding maximum and turning speed during running “suggested that VI players ran approximately 30% slower when compared to SIG [sighted players]”, and that “trunk flexion angle at the lumbar” and “thoracic levels” were significantly greater for VI in comparison to both BFO [blindfolded players] and SIG […].
Therefore, we can infer that running-related injuries (RRI) “may result from accumulated microtrauma caused by combinations of high load magnitudes (vertical ground reaction forces; vGRFs) and numbers (strides)” [6] are more distributed in Fa5 players, especially in the upper body as also evidenced by Gámez-Calvo, L.; Muñoz-Jiménez, J.; Rojas-Valverde, D. in this paper under review at lines 235-241.
…………………………………………..
REQUEST OF AMENDMENTS:
1. Please check the word “questionaire” at line 136 of the resubmitted version: “wellness validated questionaire review sheet based…”,
2. Please check the adjective “in jured” at line 159 of the resubmitted version: “for the match in jured players by match…”,
3. Please redo a couple of Tables with all data from which you draw your MANOVA and put it in Appendix at the end of paper to show how many sets of variables you have taken into account to infer your own conclusions.
4. Should you like to get some Table from me, I am here to help together with the Assistant Editor at MDPI.
With Kindest Regards,
…………………………………………..
References:
[1] National Academies of Sciences, Engineering, and Medicine. 2018. Improving Health Research on Small Populations: Proceedings of a Workshop. Washington, DC: The National Academies Press. https://doi.org/10.17226/25112
[2] McLaughlin JF, Winker K. An empirical examination of sample size effects on population demographic estimates in birds using single nucleotide polymorphism (SNP) data. PeerJ. 2020 Sep 16;8:e9939. doi: 10.7717/peerj.9939. PMID: 32995092; PMCID: PMC7501783.
[3] Georgiev G.Z., "Binomial Distribution Calculator", [online] Available at: https://www.gigacalculator.com/calculators/binomial-probability-calculator.php URL [Accessed Date: 26 Jun, 2022].
[4] Edwards WB, Taylor D, Rudolphi TJ, Gillette JC, Derrick TR. Effects of running speed on a probabilistic stress fracture model. Clin Biomech (Bristol, Avon). 2010 May;25(4):372-7. doi: 10.1016/j.clinbiomech.2010.01.001. Epub 2010 Jan 22. PMID: 20096977.
[5] Finocchietti S, Gori M, Souza Oliveira A. Kinematic Profile of Visually Impaired Football Players During Specific Sports Actions. Sci Rep. 2019 Jul 23;9(1):10660. doi: 10.1038/s41598-019-47162-z. PMID: 31337849; PMCID: PMC6650599.
[6] Kiernan D, Hawkins DA, Manoukian MAC, McKallip M, Oelsner L, Caskey CF, Coolbaugh CL. Accelerometer-based prediction of running injury in National Collegiate Athletic Association track athletes. J Biomech. 2018 May 17;73:201-209. doi: 10.1016/j.jbiomech.2018.04.001. Epub 2018 Apr 12. PMID: 29699823; PMCID: PMC6561647.
Other suggested readings (in alphabetical order):
Brown DJ, McMillan DC, Milroy R. The correlation between fatigue, physical function, the systemic inflammatory response, and psychological distress in patients with advanced lung cancer. Cancer. 2005 Jan 15;103(2):377-82. doi: 10.1002/cncr.20777. PMID: 15558809.
Butt Z, Lai JS, Rao D, Heinemann AW, Bill A, Cella D. Measurement of fatigue in cancer, stroke, and HIV using the Functional Assessment of Chronic Illness Therapy - Fatigue (FACIT-F) scale. J Psychosom Res. 2013 Jan;74(1):64-8. doi: 10.1016/j.jpsychores.2012.10.011. Epub 2012 Nov 15. PMID: 23272990; PMCID: PMC3534851.
Cella D, Lai JS, Stone A. Self-reported fatigue: one dimension or more? Lessons from the Functional Assessment of Chronic Illness Therapy--Fatigue (FACIT-F) questionnaire. Support Care Cancer. 2011 Sep;19(9):1441-50. doi: 10.1007/s00520-010-0971-1. Epub 2010 Aug 13. PMID: 20706850.
Chandran V, Bhella S, Schentag C, Gladman DD. Functional assessment of chronic illness therapy-fatigue scale is valid in patients with psoriatic arthritis. Ann Rheum Dis. 2007 Jul;66(7):936-9. doi: 10.1136/ard.2006.065763. Epub 2007 Feb 26. PMID: 17324972; PMCID: PMC1955111.
Hagell P, Rosblom T, Pålhagen S. A Swedish version of the 16-item Parkinson fatigue scale (PFS-16). Acta Neurol Scand. 2012 Apr;125(4):288-92. doi: 10.1111/j.1600-0404.2011.01560.x. Epub 2011 Jun 21. PMID: 21692754.
Januario, L.B.; Karstad, K.; Rugulies, R.; Bergström, G.; Holtermann, A.; Hallman, D.M. Association between Psychosocial Working Conditions and Perceived Physical Exertion among Eldercare Workers: A Cross-Sectional Multilevel Analysis of Nursing Homes, Wards and Workers. Int. J. Environ. Res. Public Health 2019, 16, 3610. https://doi.org/10.3390/ijerph16193610
Kim BJ, Handcock MS. Population Size Estimation Using Multiple Respondent-Driven Sampling Surveys. J Surv Stat Methodol. 2019 Dec 7;9(1):94-120. doi: 10.1093/jssam/smz055. PMID: 33521154; PMCID: PMC7834445.
Klaver, K. M., Schagen, S. B., Kieffer, J. M., van der Beek, A. J., & Duijts, S. F. A. (2021). Trajectories of Cognitive Symptoms in Sick-Listed Cancer Survivors. Cancers, 13(10), [2444]. https://doi.org/10.3390/cancers13102444
Lahti J, Mendiguchia J, Ahtiainen J, Anula L, Kononen T, Kujala M, Matinlauri A, Peltonen V,
Thibault M, Toivonen RM, Edouard P, Morin JB (2020). Multifactorial individualised programme for hamstring muscle injury risk reduction in professional football: protocol for a prospective cohort study. BMJ Open Sport & Exercise Medicine. 6:e000758.
Lai JS, Cook K, Stone A, Beaumont J, Cella D. Classical test theory and item response theory/Rasch model to assess differences between patient-reported fatigue using 7-day and 4-week recall periods. J Clin Epidemiol. 2009 Sep;62(9):991-7. doi: 10.1016/j.jclinepi.2008.10.007. Epub 2009 Feb 12. PMID: 19216054; PMCID: PMC2771583.
Lui F, Duzzi D, Corradini M, Serafini M, Baraldi P, Porro CA. Touch or pain? Spatio-temporal patterns of cortical fMRI activity following brief mechanical stimuli. Pain. 2008 Aug 31;138(2):362-374. doi: 10.1016/j.pain.2008.01.010. Epub 2008 Mar 4. PMID: 18313223.
Majumdar, A., Bakirov, R., Hodges, D. et al. Machine Learning for Understanding and Predicting Injuries in Football. Sports Med - Open 8, 73 (2022). https://doi.org/10.1186/s40798-022-00465-4
Nilsson MH, Bladh S, Hagell P. Fatigue in Parkinson's Disease: Measurement Properties of a Generic and a Condition-specific Rating Scale. J Pain Symptom Manage. 2013 Mar 15. pii: S0885-3924(13)00107-3. doi: 10.1016/j.jpainsymman.2012.11.004. [Epub ahead of print]
Okifuji, A., & Hare, B. D. (2015). The association between chronic pain and obesity. Journal of pain research, 8, 399–408. https://doi.org/10.2147/JPR.S55598
Smith E, Lai JS, Cella D. Building a measure of fatigue: the functional assessment of Chronic Illness Therapy Fatigue Scale. PM R. 2010 May;2(5):359-63. doi: 10.1016/j.pmrj.2010.04.017. PMID: 20656617.
Yost KJ, Eton DT. Combining distribution- and anchor-based approaches to determine minimally important differences: the FACIT experience. Eval Health Prof. 2005 Jun;28(2):172-91. doi: 10.1177/0163278705275340. PMID: 15851772.
Wilkerson GB, Nabhan DC, Perry TS. A Novel Approach to Assessment of Perceptual-Motor Efficiency and Training-Induced Improvement in the Performance Capabilities of Elite Athletes. Front Sports Act Living. 2021 Oct 1;3:729729. doi: 10.3389/fspor.2021.729729. PMID: 34661098; PMCID: PMC8517233.

Author Response
Dear Editor and reviewers:
We have carefully considered all reviewers' recommendations for the paper (Manuscript ID: ijerph-1796805) entitled "Analysis of injuries and well-being in blind athletes during an international football competition”. Please find enclosed our detailed answers to reviewers' queries. The authors declare that the manuscript is original and has not been considered for publication elsewhere. Additionally, the authors had approved the paper for release and agree with its content.
Please find all corrections in red inside the manuscript.
Reviewer 1
R1.1.The reviewers' recommendations were followed or clarified according to the request. The final document presents robust and well-developed data on the topic and the reviews covered the gaps found in the review.
R/As Corresponding Author On Behalf Of All Authors We Thank The Reviewers For Their Contributions To Improve The Quality Of This MS.
R1.2. Introduction:
Describes important points with an updated reference on the topic.
R/We really thank the reviewer for this consideration.
R1.3. Materials and methods:
Clearly defines the way in which the work is carried out.
R/We want to thank the reviewer for his/her insights.
R1.4. Results:
I believe that table 2 can be replaced by a small paragraph.
Robust data is presented clearly and objectively.
R/We think the table 2 its needed to clarify which player and when did the player get injured. We want to request the reviewer to maintain the table.
R1.5.Discussion:
Even if it is short, it presents data that collaborate with the research, in this way, well elaborated.
R/Thank you very much for your comments.
R1.6. Conclusion:
Objective with the results and presenting the limitations directly.
R/The limitation, was added as a new section considering additional information.
Reviewer 2
R2.1. The presented manuscript has an interesting premise and analyses the issue of injury incidence among 5-a-side soccer players, which is interesting from the point of view of disabled sports. It fits the profile of the IJERPH journal, including Special Issue: Data Analytics and Statistical Approaches Applied in Injury Risk, Illness, Well-Being, and Exercise Monitoring.
R/ As Corresponding Author On Behalf Of All Authors We Thank The Reviewers For Their Contributions To Improve The Quality Of This MS
In order for it to be published, I ask for clarification and completion of important points:
R2.2. You can provide the approval number of the relevant Research Ethics Committee in the abstract.
R/We have added the registration code of the REC or IRB in the text. The Institutional Review Board (Universidad de Extremadura, Reg. Code 67/2017).
R2.3. The questionnaire on the well-being of the subjects should be described in more detail.
R/More details regarding the mood/wellness questionnaire were added. More clarification was given on the validation, interpretation and protocols.
R2.4. You should state what tests were used to determine whether the data obtained were parametric or non-parametric.
R/Thank you for the opportunity to clarify this issue. We have stated the normality test used.
R2.5. Include a separate section on the limitation of the study! This is partially described in the conclusions, but with this size of the study group, this section seems a necessity.
R/The limitation was added as a separate section and it was completed with new considerations.
Reviewer 3
On behalf of the authors as corresponding author, we must thank the time and scientific criteria with which the reviewer has analyzed this scientific article. In this sense, we have incorporated, considering our knowledge and our possibilities, as well as other corrections from reviewers 1 and 2; the recommendations made.
Consequently, we have incorporated those recommendations that allow other reviews to be respected, but we ask the reviewer to open the door for prompt collaborations for the best use of the data obtained.
R3.1. Please check the word “questionaire” at line 136 of the resubmitted version: “wellness validated questionaire review sheet based…”
R/The word was corrected.
R3.2. Please check the adjective “in jured” at line 159 of the resubmitted version: “for the match in jured players by match…”,
R/ The word was corrected.
R3.3. Please redo a couple of Tables with all data from which you draw your MANOVA and put it in Appendix at the end of paper to show how many sets of variables you have taken into account to infer your own conclusions.
R/
R3.4. Should you like to get some Table from me, I am here to help together with the Assistant Editor at MDPI.
R/ We definitely consider it very valuable to collaborate with reviewer 3 in the sense of taking advantage of the information. The approach given is very interesting and we consider the approach to the data that you have made to be very valuable. The techniques carried out are not our domain, but we open the door for the approach. We believe that for this, the review must be more open, which the MDPI system does not allow us to know if it allows. But, we want to be able to have the contact of an open review to exchange this knowledge to enrich the results.
We have incorporated everything that is within our resources.

Reviewer 2 Report
The presented manuscript has an interesting premise and analyzes the issue of injury incidence among 5-a-side soccer players, which is interesting from the point of view of disabled sports. It fits the profile of the IJERPH journal, including Special Issue: Data Analytics and Statistical Approaches Applied in Injury Risk, Illness, Well-Being, and Exercise Monitoring.
In order for it to be published, I ask for clarification and completion of important points:
1. You can provide the approval number of the relevant Research Ethics Committee in the abstract.
2. The questionnaire on the well-being of the subjects should be described in more detail.
3. You should state what tests were used to determine whether the data obtained were parametric or non-parametric.
4. Include a separate section on the limitation of the study! This is partially described in the conclusions, but with this size of the study group, this section seems a necessity.
Author Response

(The authors gave the same response as above.)

Reviewer 3 Report
The reviewers' recommendations were followed or clarified according to the request. The final document presents robust and well-developed data on the topic and the reviews covered the gaps found in the review.
Introduction:
Describes important points with an updated reference on the topic.
Materials and methods:
Clearly defines the way in which the work is carried out.
Results:
I believe that table 2 can be replaced by a small paragraph.
Robust data presented clearly and objectively.
Discussion:
Even if it is short, it presents data that collaborate with the research, in this way, well elaborated.
Conclusion:
Objective with the results and presenting the limitations directly.
Author Response
Dear Editor and reviewers:
We have carefully considered all reviewers' recommendations for the paper (Manuscript ID: ijerph-1796805) entitled "Analysis of injuries and well-being in blind athletes during an international football competition”. Please find enclosed our detailed answers to reviewers' queries. The authors declare that the manuscript is original and has not been considered for publication elsewhere. Additionally, the authors had approved the paper for release and agree with its content.
Please find all corrections in red inside the manuscript.
Reviewer 1
R1.1.The reviewers' recommendations were followed or clarified according to the request. The final document presents robust and well-developed data on the topic and the reviews covered the gaps found in the review.
R/As Corresponding Author On Behalf Of All Authors We Thank The Reviewers For Their Contributions To Improve The Quality Of This MS.
R1.2. Introduction:
Describes important points with an updated reference on the topic.
R/We really thank the reviewer for this consideration.
R1.3. Materials and methods:
Clearly defines the way in which the work is carried out.
R/We want to thank the reviewer for his/her insights.
R1.4. Results:
I believe that table 2 can be replaced by a small paragraph.
Robust data is presented clearly and objectively.
R/We think the table 2 its needed to clarify which player and when did the player get injured. We want to request the reviewer to maintain the table.
R1.5.Discussion:
Even if it is short, it presents data that collaborate with the research, in this way, well elaborated.
R/Thank you very much for your comments.
R1.6. Conclusion:
Objective with the results and presenting the limitations directly.
R/The limitation, was added as a new section considering additional information.
Reviewer 2
R2.1. The presented manuscript has an interesting premise and analyses the issue of injury incidence among 5-a-side soccer players, which is interesting from the point of view of disabled sports. It fits the profile of the IJERPH journal, including Special Issue: Data Analytics and Statistical Approaches Applied in Injury Risk, Illness, Well-Being, and Exercise Monitoring.
R/ As Corresponding Author On Behalf Of All Authors We Thank The Reviewers For Their Contributions To Improve The Quality Of This MS
In order for it to be published, I ask for clarification and completion of important points:
R2.2. You can provide the approval number of the relevant Research Ethics Committee in the abstract.
R/We have added the registration code of the REC or IRB in the text. The Institutional Review Board (Universidad de Extremadura, Reg. Code 67/2017).
R2.3. The questionnaire on the well-being of the subjects should be described in more detail.
R/More details regarding the mood/wellness questionnaire were added. More clarification was given on the validation, interpretation and protocols.
R2.4. You should state what tests were used to determine whether the data obtained were parametric or non-parametric.
R/Thank you for the opportunity to clarify this issue. We have stated the normality test used.
R2.5. Include a separate section on the limitation of the study! This is partially described in the conclusions, but with this size of the study group, this section seems a necessity.
R/The limitation was added as a separate section and it was completed with new considerations.
Reviewer 3
On behalf of the authors as corresponding author, we must thank the time and scientific criteria with which the reviewer has analyzed this scientific article. In this sense, we have incorporated, considering our knowledge and our possibilities, as well as other corrections from reviewers 1 and 2; the recommendations made.
Consequently, we have incorporated those recommendations that allow other reviews to be respected, but we ask the reviewer to open the door for prompt collaborations for the best use of the data obtained.
R3.1. Please check the word “questionaire” at line 136 of the resubmitted version: “wellness validated questionaire review sheet based…”
R/The word was corrected.
R3.2. Please check the adjective “in jured” at line 159 of the resubmitted version: “for the match in jured players by match…”,
R/ The word was corrected.
R3.3. Please redo a couple of Tables with all data from which you draw your MANOVA and put it in Appendix at the end of paper to show how many sets of variables you have taken into account to infer your own conclusions.
R/
R3.4. Should you like to get some Table from me, I am here to help together with the Assistant Editor at MDPI.
R/ We definitely consider it very valuable to collaborate with reviewer 3 in the sense of taking advantage of the information. The approach given is very interesting and we consider the approach to the data that you have made to be very valuable. The techniques carried out are not our domain, but we open the door for the approach. We believe that for this, the review must be more open, which the MDPI system does not allow us to know if it allows. But, we want to be able to have the contact of an open review to exchange this knowledge to enrich the results.
We have incorporated everything that is within our resources.

This manuscript is a resubmission of an earlier submission. The following is a list of the peer review reports and author responses from that submission.
Round 1
Reviewer 1 Report
The objective of this article is to provide insights in addressing Fa5 collaborative-opposition sport, a paralympic 5-a-side football for blind competitors. Given the lack of attention paid to the perception of pain and its importance for athletes’ daily life and physical performance, the authors mean to analyze the association between space-time and mechanical load with injuries and well-being in blind players during an international Fa5 competition. See lines 78-81.
Inquired association: space-time and mechanical load with injuries / well-being and injury incidence of players (blind paralympic players).
To investigate the association between pain perception, spatio-temporal, mechanical and metabolic workload with injuries—given that this kind of sport involves numerous impacts—the authors have monitored during an international Fa5 tournament the following variables:
- general well-being,
- perception of pain and injuries, and
- the spatio-temporal, mechanical and metabolic workload.
See the abstract, lines 14-21. The authors hold permission to this research by the International Paralympic Committee (IPC) and the Institutional Review Board (Universidad de Extremadura, Reg. Code 67/2017), other than by the same team under examination. See a little forwards, lines 106-109.
In their 1. Introduction, the authors introduce basics on football, a sport up to date with a more significant number of adapted modalities respect to other playing competitions. Fa5, a five-a-side football for paralympic athletes is authorized by the FIFA (International Federation of Associated Football) since Athens 2004. A description of its features for competitors is shown at lines 39-46. The authors enhance how Fa5 is not really inquired in the scientific literature for now, and therefore they highlight how it would be a beautiful example to quantify the external loads of athletes through the information provided about the magnitude and frequency of actions based on accelerometry and actions based on speed. See specifically lines 47-56. Other variables are generally inquired in this kind of studies and related to indicators of internal loads, such as the heart rate.
In section 2. Materials and Methods, the authors exemplify their pool of reference: ten blind football players of the Spanish senior national team monitored during the Blind Football International Tournament held in Sevilla, Spain, in 2019. Spatio-temporal, mechanical and metabolic workload variables were clustered using exploratory factor analysis, while the association between the grouped (factors) variables was associated with players' wellness and injury incidence. See lines 93-96, and Table 1 at line 92 which recaps the chronological order of the matches played by the Spanish team. Table 2 at line 103 recaps the main descriptive features of the ten players, among which (i) date of birth, (ii) height, (iii) weight, (iv) position, and (v) playing time by match (three matches). Players in the pool of reference have been placed only when playing at least 20 minutes that is the 2/3 of the match among the three considered matches, therefore only the activity when the players were participating in the game was considered. See lines 104-105.
In sub-paragraphs 2.3 Materials and procedures, the authors mention in sub 2.3.1 the commercial inertial sensor has recorded mechanical data. Devices were placed in the inter-scapulae line of players, together with a heart rate metabolic indicator. The authors have made use of the same inertial measurement units WIMU PROTM (RealTrack Systems, Almeria, Spain) as by Gómez-Carmona, C.D.; Pino-Ortega, J.; et al. (2019). Gómez-Carmona, C.D.; Pino-Ortega, J.; et al. (2019) have analyzed the agreement among the different accelerometry-based load indicators available in sport science. In fact, static high-intensity actions without covering ground (jumps, collisions, falls, tackles, etc.) cannot be recorded by time-motion systems but can be measured with high accuracy by accelerometers [see: Gómez-Carmona, C.D.; Pino-Ortega, J.; et al. (2019)].
Collation and analysis of data from studies reporting temporal changes in the occurrence of higher intensity accelerations and decelerations during competitive match play may help acquire knowledge regarding the magnitude of the decline and potential impact that this may have on match performance and injury risk. Particularly, «intense accelerations and decelerations could be particularly vulnerable to neuromuscular fatigue and consequently to an exacerbated risk of incurring injury» [see: Harper, D.J.; Carling, C.; Kiely, J. (2019), at pag. 1924].
In sub 2.3.2 Wellness and Injury Incidence, the authors assessed the players’ general well-being, fatigue, and the presence of pain during the competition through a well-being review sheet based on athletes monitoring recommendations. See lines 124-126. A five-point Likert-scale questionnaire has scaled (i) players’ level of fatigue, (ii) sleep quality, (iii) muscle soreness, (iv) stress levels, and (v) mood with range 5 – lowest/1 – highest. Injury incidence components are shown for each of the ten players in Table 3 at line 172.
In their sub 2.3.3 Statistical Analysis, the authors have done a standard Z-score normalization and centration of absolute raw data. Because of the standardization of a universal index to calculate accelerometer load is needed in order to make possible between-study comparison, the authors have performed a two-way Analysis of Variance (ANOVA) using scores post-hoch analysis with Bonferroni’s correction. They made use also of Omega squared () to qualify the degree of association for their rather small population (ten subjects). Statistical interpolation of raw data are described in great detail at lines 134-164.
Clustering spatio-temporal and mechanical variables through an explorative PC (principal component) analysis—previously testing their sampling adequacy through a Kaiser-Meyer-Olkin (KMO) Measure and Bartlett Sphericity test significance—the authors have found very large to nearly perfect higher levels of muscle pain (ES= 0.14, large) and feeling stressed (Es: 0.4, large) correlations, while both the stress and mood gradually decreased as the tournament progressed. See section 3. Results at line 166 forwards and see also Table 4 at line 180. According to authors, their PCA provides then key information about factors that explain the variance of the spatio-temporal variables (Distance, Explosive Distance, Maximum Speed and Average Speed) and the mechanical variables of workload (Accelerations, Decelerations, Impacts, Steps, and Player Load). See a little forwards at lines 230-233.
There were no differences in Spatio-temporal variables by match and injury incidence, as shown in Table 5 at line 184. Instead, very large differences were found in scaled and centered values in mechanical workload variables, as shown in Table 6 at line 190. The authors have commented their PC using of a binomial logistic regression through a backward elimination method. See specifically lines 159-164.
To interpret each principal components, the authors have examined the magnitude and direction of the coefficients for the original variables in Figure 1 at line 192. The larger the absolute value of the coefficient speed and distance, the more important the corresponding variable is in calculating the component. While the mechanical workload variables explained lowly the incidence pf injury, perceived muscle soreness and stress level explained injury incidence significantly. See lines 198-204.
In their section 4. Discussion, the authors strengthen how Fa5 confirms itself a paralympic sport with the highest incidence of injuries of all those analyzed, with injuries especially in the upper body and lesions in the lower limbs. See lines 206-217. Pain and the perception of pain result to be highly linked to sports experience and performance, then they negatively affect the performance of athletes.
In fact, the authors find further large differences in pain perception and stress levels among the injured players, who presented higher levels of muscle pain and stress than the non-injured players. Values are anyway higher than expected, also considering that in a non-paralympic football match it is more likely workload resulting from lumbar segments will tend to be higher than those assessed in thoracic segments. See lines 206-248.
To the lack of information in the sport science area related to sampling rates, chip sets, filtering methods and data-processing algorithms, especially in Fa5, which makes it impossible to compare devices from the accelerometer raw data, the authors advance against that their inquiry ease the fact players’ internal and external load did not explain the injury incidence. This is contravened by the values obtained in the well-being questionnaire, with the perception of pain and stress as contributing factors. It is therefore explicated in the section 5. Conclusions this is an experimental paper due to the lack of scientific articles published on Fa5, with the strong limitation the sample is rather small, and data are someway indexed along the three collecting different matches’ times for data on the same ten players.
MAJOR CHANGE REQUEST:
- Please provide a complete definition of injury to the audience in your understanding somewhere at line 131,
- Since a good vision and sensorimotor skills are necessary to either avoid impacts or prepare the body to reduce among others the severity of head impacts and head kinematics, please explain at least in the “Conclusions” how your paper may improve the blind athletes’ ability to anticipate collisions and thus brace for impact to reduce the magnitude of head accelerations,
- Since your research is a kind of meta- Time-To-Event (TTE) analysis, consider the “recall bias” as a factor to weight in final results. You could or get more information on injuries than in need because of the interest of interviewees to fill the survey, or less than effectively occurred because the blind players do not assess appropriately the intensity of stressful events.
This way you could add a proportional well-being and injury incidence of players assumption, based on specific disability weights (DWs) derived from blind patient experiences and hospital records.
CHANGE REQUEST:
- Please add “load indexes,” and “sport technology” to keywords at line 26,
- Please solve the acronym ANOVA (Analysis of Variance) at line 143 for non-practitioners of your study area,
- I think in “wellness scores post-Hoch analysis,” at line 145, “hoc” should be written with small initial letter. Please check with the Editor,
- Please solve the acronym KMO (Kaiser-Meyer-Olkin) values at line 152 for non-practitioners of your study area,
- Please quote whether possible the MDPI paper by Gómez-Carmona, C.D.; Pino-Ortega, J.; et al. (2019) is at the bottom of this report,
With Kind Regards,
References:
Finocchietti, S., Gori, M. & Souza Oliveira, A. Kinematic Profile of Visually Impaired Football Players During Specific Sports Actions. Sci Rep, 2019, 9, 10660. https://doi.org/10.1038/s41598-019-47162-z
Harper, D.J.; Carling, C.; Kiely, J. High-Intensity Acceleration and Deceleration Demands in Elite Team Sports Competitive Match Play: A Systematic Review and Meta-Analysis of Observational Studies. Sports Med, 2019, 49, 1923–1947. https://doi.org/10.1007/s40279-019-01170-1. Available from: https://link.springer.com/content/pdf/10.1007/s40279-019-01170-1.pdf (accessed on March 13, 2022).
Gómez-Carmona, C.D.; Pino-Ortega, J.; Sánchez-Ureña, B.; Ibáñez, S.J.; Rojas-Valverde, D. Accelerometry-Based External Load Indicators in Sport: Too Many Options, Same Practical Outcome? Int. J. Environ. Res. Public Health 2019, 16, 5101. https://doi.org/10.3390/ijerph16245101
Kok, M.; Hol, J.D.; Schön, T.B. Using Inertial Sensors for Position and
Orientation Estimation. Foundations and Trends in Signal Processing 2017, 11, 1-2, 1-153. http://dx.doi.org/10.1561/2000000094
Kung, S.M., Suksreephaisan, T.K., Perry, B. et al. The Effects of Anticipation and Visual and Sensory Performance on Concussion Risk in Sport: A Review. Sports Med – Open, 2020, 6, 54 (2020). https://doi.org/10.1186/s40798-020-00283-6
Pino-Ortega, J., Rojas-Valverde, D., Gómez-Carmona, C. D., & Rico-González, M. Training Design, Performance Analysis, and Talent Identification-A Systematic Review about the Most Relevant Variables through the Principal Component Analysis in Soccer, Basketball, and Rugby. International journal of environmental research and public health, 2021, 18(5), 2642. https://doi.org/10.3390/ijerph18052642
Sprouse, B., Alty, J., Kemp, S. et al. The Football Association Injury and Illness Surveillance Study: The Incidence, Burden and Severity of Injuries and Illness in Men’s and Women’s International Football. Sports Med, 2020. https://doi.org/10.1007/s40279-020-01411-8
Tung, K.T.S., Ho, F.K., Wong, W.H.S. et al. Quantification of injury burden using multiple data sources: a longitudinal study. Sci Rep, 2021, 11, 3078. https://doi.org/10.1038/s41598-021-82799-9
Udby C.L., Impellizzeri F.M., Lind M., Nielsen R.Ø. How Has Workload Been Defined and How Many Workload-Related Exposures to Injury Are Included in Published Sports Injury Articles? A Scoping Review. J Orthop Sports Phys Ther. 2020 Oct;50(10):538-548. doi: 10.2519/jospt.2020.9766. PMID: 32998614.

Author Response
Dear, reviewer.
Dear, Reviewer.
I enclose in a pdf file the corrections of the document.
Greeting

Reviewer 2 Report
Dear Authors,
your study aimed to define which factors could be associated with injury risk in five-a-side football.
I think the topic is interesting as if many injuries happen in this sport, is important to understand the underlying factors in order to reduce injury incidence.
The introduction is well written, even if several English and grammar errors are present throughout the manuscript. I suggest a careful revision. For example, I think the first three lines after the heading "Results" are oversight of the formatting form. Please, check.
I have some concerns regarding the methods and the presentation of your results.
First of all, as you might expect, the first issue is the sample size. 10 participants are too low to make inferences over a population and to explain a multifactorial issue such as injury risk.
Another important point is the definition of "injury". Is contusion considered an injury in similar studies or other disciplines? I mean, 100% of the players that get injured (excluding the goalkeepers) seems a very severe problem! So, I question the definition of "injury" you use, because this is what your all your manuscript stands.
I am not totally convinced by your statistical analysis. Were the repeated measures considered in the ANOVA? Please, provide a better explanation of which variables were used as independent and dependent variables. In the tables you reported an "injury effect", but I do not understand what you mean. Moreover, you reported n=16 injured and n=14 non-injured, but you only have 10 participants, so you can't sum up injuries as they were different persons.
I suggest removing table 1 as it is useless and removing birth dates from table 2, it is unusual to report them. The mean age is sufficient as reported in the text.
Was the questionnaire you used for the wellbeing variables validated? If yes, please state it. Moreover, please be consistent in the use of terms such as wellness or wellbeing.
I do not understand why Distance is reported in (meters/minute) in table 5. Meters/minute is a measure of speed, not distance. Was the raw distance (meters) used for the analyses?
Finally, please consider a broader discussion of the reasons why the factors you measured would contribute to increasing the risk of concussion.
I look forward to receiving the revised version of your manuscript.
Best regards
Author Response
General comments
Open Review
(x) I would not like to sign my review report
( ) I would like to sign my review report
English language and style
( ) Extensive editing of English language and style required
(x) Moderate English changes required
( ) English language and style are fine/minor spell check required
( ) I don't feel qualified to judge about the English language and style
The authors really appreciate all your kindly comments related to our manuscript during the peer-review process that has contributed to improve the readability and the quality of the present manuscript.
|
Yes |
Can be improved |
Must be improved |
Not applicable |
|
|
Does the introduction provide sufficient background and include all relevant references? |
( ) |
(x) |
( ) |
( ) |
|
Is the research design appropriate? |
( ) |
(x) |
( ) |
( ) |
|
Are the methods adequately described? |
( ) |
( ) |
(x) |
( ) |
|
Are the results clearly presented? |
( ) |
( ) |
(x) |
( ) |
|
Are the conclusions supported by the results? |
( ) |
( ) |
(x) |
( ) |
Authors thanks the general comments provided by the reviewer.
Comments and Suggestions for Authors
Dear Authors,
your study aimed to define which factors could be associated with injury risk in five-a-side football.
I think the topic is interesting as if many injuries happen in this sport, is important to understand the underlying factors in order to reduce injury incidence.
Thank you for your feedback.
The introduction is well written, even if several English and grammar errors are present throughout the manuscript. I suggest a careful revision. For example, I think the first three lines after the heading "Results" are oversight of the formatting form. Please, check.
Thank you for your feedback. In the new document, the English has been corrected.
I have some concerns regarding the methods and the presentation of your results.
First of all, as you might expect, the first issue is the sample size. 10 participants are too low to make inferences over a population and to explain a multifactorial issue such as injury risk.
Another important point is the definition of "injury". Is contusion considered an injury in similar studies or other disciplines? I mean, 100% of the players that get injured (excluding the goalkeepers) seems a very severe problem! So, I question the definition of "injury" you use, because this is what your all your manuscript stands.
Thank you very much. The document has been improved. Also, a more exhaustive definition of injury has been given. Regarding the participants, it is the total number of players, and it is a sample of quality since they are international Paralympic athletes, and those who have competed in the last Paralympic Games in Tokyo.
I am not totally convinced by your statistical analysis. Were the repeated measures considered in the ANOVA? Please, provide a better explanation of which variables were used as independent and dependent variables. In the tables you reported an "injury effect", but I do not understand what you mean. Moreover, you reported n=16 injured and n=14 non-injured, but you only have 10 participants, so you can't sum up injuries as they were different persons.
Thank you very much for your comment. In the new version, it has been improved and explained that they are from three soccer games.
I suggest removing table 1 as it is useless and removing birth dates from table 2, it is unusual to report them. The mean age is sufficient as reported in the text.
Thanks for your comment. The date of birth has been removed, and the Body Mass Index has been incorporated.
Was the questionnaire you used for the wellbeing variables validated? If yes, please state it. Moreover, please be consistent in the use of terms such as wellness or wellbeing.
Thank you very much for the comment. The questionnaire is validated and is used in research related to high performance sports.
I do not understand why Distance is reported in (meters/minute) in table 5. Meters/minute is a measure of speed, not distance. Was the raw distance (meters) used for the analyses?
Thank you. It refers to the explosive distance.
Finally, please consider a broader discussion of the reasons why the factors you measured would contribute to increasing the risk of concussion.
I look forward to receiving the revised version of your manuscript.
Best regards
Thank you for your feedback. In the new document, the mistakes have been improved.

Round 2
Reviewer 1 Report
Dear Authors,
Thank you so much for having taken so seriously my suggestions and comments. The paper is improved and demonstrates a strong commitment to research and data.
With Kindest Regards,
Reviewer 2 Report
Dear Authors,
I'll be honest with you. I'm not satisfied at all with the attention you paid in doing the revision. Seems that you provided me generic answers without really taking into consideration my comments.
I asked to revise the English form, but several errors are still present (for example, look at the caption of Table 2). I asked a better explanation of how you conducted the ANOVA, but it does not seem to be present. Further you are still considering the people "injured" and "non-injured" as different people in the three matches and not as paired data (it is not possible to have 16 injured people if the sample is 10. You have people who got injured once or twice, and this data should be treated appropriately). The first three lines of the results are still present (obviously an error), and as I highlighted it before and you seem to have ignored it, I regret to suggest a rejection this time.